# Investigation of the Presence of DNA in Human Blood Plasma Small Extracellular Vesicles

**DOI:** 10.3390/ijms24065915

**Published:** 2023-03-21

**Authors:** Kristína Lichá, Michal Pastorek, Gabriela Repiská, Peter Celec, Barbora Konečná

**Affiliations:** 1Institute of Molecular Biomedicine, Faculty of Medicine, Comenius University in Bratislava, 811 08 Bratislava, Slovakia; 2Institute of Physiology, Faculty of Medicine, Comenius University in Bratislava, 813 72 Bratislava, Slovakia; 3Institute of Pathophysiology, Faculty of Medicine, Comenius University in Bratislava, 811 08 Bratislava, Slovakia

**Keywords:** exosomes, nuclear DNA, mitochondrial DNA, DNase, protected DNA

## Abstract

Extracellular DNA (ecDNA) is DNA outside of cells, which is a result of various mechanisms. EcDNA is believed to be a cause of various pathogeneses as well as their potential biomarker. EcDNA is believed to also be part of small extracellular vesicles (sEVs) from cell cultures. If ecDNA is present in sEVs in plasma, their membrane may protect it from degradation by deoxyribonucleases. Moreover, sEVs play a role in the intercellular communication, and they can therefore transfer ecDNA between cells. The aim of this study was to investigate the presence of ecDNA in sEVs isolated from fresh human plasma by the ultracentrifugation and density gradient, which serves to exclude the co-isolation of non-sEVs compartments. The novelty of the current study is the investigation of the localization and subcellular origin of the ecDNA associated with sEVs in plasma, as well as the estimation of the approximate concentration. The cup-shaped sEVs were confirmed by transmission electron microscopy. The highest concentration of particles was in the size of 123 nm. The presence of the sEVs markers CD9 and TSG101 was confirmed by western blot. It was found that 60–75% of DNA is on the surface of sEVs, but a part of the DNA is localized inside the sEVs. Moreover, both nuclear and mitochondrial DNA were present in plasma EVs. Further studies should focus on the potential harmful autoimmune effect of DNA carried by plasma EVs or specifically sEVs.

## 1. Introduction

The discovery of extracellular DNA (ecDNA) by Mandel and Metais [1] caused a major turning point in non-invasive diagnostics. It is DNA outside the cells that originates either from the nucleus or from the mitochondria, and it is found free in the circulation. EcDNA can be released into the circulation by several mechanisms, such as the processes of apoptosis, necrosis, oncosis or pyroptosis or the formation of neutrophil extracellular traps, NETosis [2,3]. EcDNA can be degraded by deoxyribonucleases (DNases) active in the circulation, and it was already hypothesized that it can be protected from nuclease digestion [4,5]. EcDNA could have an immunomodulatory effect [4,5,6] and proinflammatory effect [5], and it could be involved in the development of various diseases, such as autoimmune diseases [7], inflammatory bowel diseases [8] or cardiovascular diseases [9,10]. EcDNA could be in the circulation in various forms—free or as a component of various structures, such as nucleosomes [11], virtosomes [12], neutrophil extracellular traps [13] or extracellular vesicles (EVs) [7].

Small EVs (sEVs) are a subset of EVs which have spherical shape and range from 30 to 150 nm in size [14]. They are released in vivo by all cells of the body [15] and in vitro by being cultivated into the cell medium [16]. They are found in all body fluids—blood, saliva, urine, bronchoalveolar fluid, cerebrospinal fluid, breast milk, amniotic fluid, synovial fluid, tears, lymph, bile or gastric acid [17]. sEVs are composed of a lipid membrane, proteins, nucleic acids and metabolites. Some proteins are derived from cells and tissues from which vesicles have been released. Other proteins are thought to be specific sEVs proteins such as tetraspanins CD9, CD63, CD81, proteins of the ESCRT complex and its other proteins, such as Alix, TSG101 and HSP70, involved in the biogenesis of sEVs [14,18].

Various studies show different contradictory results on the EVs-associated DNA, which might be related to the methodology process. Fernando et al. (2017) have demonstrated that approximately 90% of ecDNA in human plasma is found in exosomes [19]. Since ecDNA is found in sEVs, it is protected from cleavage by DNases. However, Jeppesen et al. (2019) published a different study with a strongly supported conclusion that DNA is not found in exosomes, and the authors suggest a reassessment of the composition of exosomes [20]. The authors proposed a new model of the active secretion of ecDNA by a mechanism that is dependent on autophagy and the multivesicular body involved in exosome biogenesis. This proposed model is independent of the transport of DNA by exosomes. Exosomes were, however, isolated from the human colon (DKO-1) and glioblastoma (Gli36) cancer cell line. Yokoi et al. [21] showed that cancer cell lines produce exosomes with a high content of nuclear DNA (ncDNA) and proposed the mechanisms of loading such DNA into exosomes. Repiská et al. [22] suggest that not only maternal DNA but also fetal DNA are associated with plasma exosome-like vesicles. It is also reported that not only sEVs but also other EVs contain DNA. It seems that larger vesicles contain more complex DNA [23]. Lazo et al. suggested that EVs from human plasma are associated with mitochondrial DNA (mtDNA), which declines with age [24]. More evidence was reported on the topic of oncology and EVs-associated DNA [25,26]; however, isolation procedures and the starting material differ considerably. Given the published findings, the presence and localization of DNA associated with EVs, especially sEVs, are still unclear and discussed in the EV research field.

The aim of this study was to investigate the presence of ecDNA in sEVs isolated from fresh human plasma by the isolation method inspired by the article of Jeppesen et al. (2019). This method of isolation serves to exclude the false positive results caused by the co-isolation of non-EV compartments. The novelty of the current study is to clearly investigate the localization and subcellular origin of the ecDNA associated with sEVs as well as the estimation of the approximate concentration.

## 2. Results

### 2.1. Confirmation of Isolated sEVs

The presence of sEVs was confirmed by transmission electron microscopy (Figure 1A), western blot (Figure 1B) and nanoparticle tracking analysis (Figure 2). The shape and the size of isolated vesicles corresponded to that of sEVs (Figure 1A). The presence of the 44 kDa TSG101 marker and the 21 kDa CD9 marker was confirmed by western blot. The presence of 90 kDa Calnexin as a negative control was not confirmed (Figure 1B).

The size profile of sEVs was in the range of 40–130 nm, while the highest concentration of the isolated particles had a size of 123 nm (Figure 2A). The average particle size was 150 nm. The particle concentration was 0.88 × 10^8^ particles/mL of plasma. The size profile of the sEVs isolated from DNase-treated plasma (Dplas-sEVs) particles is in the range of 16–40 nm, while in the highest concentration in the isolated samples, there are particles with a size of 28 nm (Figure 2B). The average particle size is 36 nm. The concentration of particles is 3.55 × 10^12^ particles/mL plasma. The resulting size profile of sEVs treated with DNase (D-sEVs) is in the range of 22–37 nm, while in the highest concentration in the isolated samples, there are particles with a size of 26 nm (Figure 2C). The average particle size is 62 nm. The concentration of particles is 2.76 × 10^12^ particles/mL plasma.

### 2.2. Presence of DNA in sEVs

The fluorescence signal of particles stained with Sytox Green after DNase treatment decreases (Figure 3B,E), as well as the number of particles in the sample (Figure 3D), in comparison to untreated samples (Figure 3A,D,E). The histogram shows the distribution of Sytox-stained particles in a representative sample without DNase treatment and with DNase treatment (Figure 3C).

### 2.3. Concentration of ecDNA in sEVs

EcDNA concentrations were measured in plasma and in sEVs (Figure 4A). The plasma ecDNA concentration was 23.11 ± 6.96 ng/mL of plasma. The ecDNA concentration in sEVs was 0.47 ± 0.09 ng/mL of plasma. The ecDNA concentration in Dplas-sEVs was 0.67 ± 0.16 ng/mL of plasma. The ecDNA concentration in D-sEVs was 0.21 ± 0.07 ng/mL of plasma. The resulting concentrations are the mean of the measured results ± SEM.

### 2.4. Protected or Unprotected DNA

Measured relative fluorescence units were normalized to the percentage, where 100% was the highest measured fluorescence signal in the samples treated with Triton X–100 (Figure 4B). After comparing the fluorescence signal in the samples, 60–75% of the ecDNA is located on the surface of sEVs. The values are the percentage of relative fluorescence units ± SEM.

### 2.5. Subcellular Origin of ecDNA

NcDNA was detected in all plasma samples, but it was not detected in all sEVs samples with or without DNase treatment. The NcDNA in plasma was 12,635 ± 4353 copies/mL of plasma. The NcDNA in sEVs was 3 ± 2 copies/mL of plasma; in Dplas-sEVs, it was 4 ± 3 copies/mL of plasma; in D-sEVs, it was 2 ± 1 copies/mL of plasma (Figure 5A). MtDNA was detected in all plasma samples and in all samples of sEVs. The MtDNA in plasma was 261,517,644 ± 107,518,264 copies/mL of plasma. The MtDNA in sEVs was 218 ± 42 copies/mL of plasma; in Dplas-sEVs, it was 258 ± 77; the mtDNA in D-sEVs was 255 ± 95 copies/mL of plasma (Figure 5B).

## 3. Discussion

The question of the origin, location and concentration of DNA present in plasma sEVs is still unanswered and unclear from already published, contradictory studies. Therefore, in this study, we first tried to isolate sEVs from human plasma by iodixanol density gradient ultracentrifugation (UC), the specific procedure inspired by Jeppesen et al. (2019), to prevent the co-isolation of non-vesicular compartments. Second, we looked at the presence of DNA and tried to find what we could confirm about it. After isolation, we validated the presence of isolated sEVs by three different methods: western blot, nanoparticle tracking analysis and transmission electron microscopy. sEVs with DNase were smaller in size, on average, compared to sEVs without DNase, according to Nanosight; however, the limitations of the instrument should be taken into consideration. We conclude that due to the addition of DNase, we were able to dissociate sEVs aggregates. These may have been formed by sticky DNA on the surface or protein structures or simply during high-speed centrifugation included in the isolation process. As Lete et al. (2014) have shown, histones could interact with negatively charged lipid membranes and induce the extensive aggregation of phosphatidylinositol-4-phosphate-containing vesicles and, to a lesser extent, phosphatidylinositol-containing vesicles. Exosomes released from astrocytes, glioblastoma cells, prostate cancer cells, reticulocytes, dendritic cells or melanoma cells contain phosphatidylinositol [27,28,29]; therefore, we propose that the interactions of histones with phosphatidylinositol could contribute to the aggregation of sEVs. Mg^2+^ cations could also be responsible for the aggregation and changes in shape of DNA, as the aggregation of DNA molecules occurred at higher concentrations of Mg^2+^ in the solution [30], which is also important to stress since blood was collected into heparin tubes in our case. Our measurements show that it is probably more appropriate to use DNase before the UC, which potentially prevents the formation of aggregates, and this, in turn, changes the size distribution profile according to our data. Shelke et al. (2016) observed an increase in the number of particles after the addition of DNase to vesicles after the isolation, which also suggests that DNA located on the surface may lead to vesicle aggregation.

The association of sEVs with ecDNA has not been sufficiently investigated in human plasma. More publications are focused on the relationship between ecDNA and sEVs isolated from cell cultures or cancer-derived sEVs. Fernando et al. (2017) demonstrated that plasma-derived exosomes contain ecDNA. We also hypothesized that sEVs from plasma could contain DNA, even though thorough research published in a reputable journal by Jeppesen et al. (2019) showed the opposite. The authors showed that exosomes do not contain DNA and that there exist other non-exosomal particles that contain DNA and are easily co-isolated with exosomes. On the other hand, different kinds of cells lines were used to isolate exosomes by Jeppesen et al. (2019). Therefore, it is difficult to conclude that the results from the study apply to sEVs in plasma. In our study, we decided to follow the step-by-step protocol in Jeppesen et al.’s study to exclude the potential isolation of non-exosomal vesicles containing DNA. Human plasma was used as a substrate in contrast to cell lines because plasma is more complex. Although being more heterogeneous, it is more reflective of the real state of the organism. According to our results, sEVs were isolated from plasma, and we confirmed the presence of ecDNA in them. Quantification showed that they contain approximately 2% of the ecDNA present in plasma. The quantitative polymerase chain reaction (qPCR) showed that most of the sEVs DNA is of mitochondrial origin.

To determine the relationship between ecDNA and sEVs, flow cytometry analysis was performed. It was observed that, after DNase treatment, the concentration of the DNA signal decreased. Since we have not observed a shift in the Sytox signal for the particular sEVs populations characterized by size but rather observed an overall decrease in their numbers, we can hypothesize that a certain portion of vesicles contain DNA protected from degradation. These results are consistent with our measurements of the concentration of DNA in the sEVs samples, where we found that the majority of ecDNA is associated with the outer membrane of sEVs. Németh et al. [31] showed that exosomes released from the Jurkat human T-cell lymphoma and U937 human histiocytic lymphoma cell lines under cyprofloxacin stimulation contain more DNA on the surface, not inside the exosomes. Liu et al. [32] also revealed that ecDNA is mainly localized on relatively small-sized EVs in comparison with large-sized EVs. However, we challenged a technical limitation, because through FC, we can observe the smallest particles around 80 nm. Considering that UC is the golden standard for the isolation of exosomes, it was surprising to see a relatively high number of particles in the size range of 100–500 nm. This could mean that larger EVs could be at least partially collected during the isolation process next to sEVs or that sEVs are clumped together as a result of UC. We can also hypothesize that, due to the fact that ecDNA is bound to the surface of sEVs and can potentially form aggregates of EVs and the size profile of the particles is in the range of 100–500 nm, DNase treatment could then free EVs from these aggregates. However, it appears that the DNA is also protected from DNase degradation on the outside of the membrane. It is possible that DNase could get inside through the non-intact membrane of particles which was created during UC [33]. Alternatively, maybe the DNase did not cleave all the ecDNA, and the shortest fragments were detectable with Sytox.

In addition to its potential use as a biomarker of diseases, ecDNA also has the ability to induce inflammatory processes in the body. Due to its similarity to bacterial DNA, mtDNA is perceived as foreign and has a greater ability to cause inflammation in the body [34]. We confirmed that mtDNA is found in sEVs and is partially protected from degradation by DNase. This can cause increased concentrations in the circulation, resulting in inflammatory processes or the exacerbation of ongoing inflammation in the body [35]. Some studies have shown that mtDNA contains unmethylated CpG motifs similar to bacterial DNA that can activate pattern recognition receptors such as TLR9, thereby activating the immune system [36]. MtDNA also has the ability to promote inflammation in the form of a cytosolic multiprotein oligomer, which activates inflammatory reactions [34]. When mice were injected with mtDNA in the joints, they developed localized inflammation and arthritis [37]. MtDNA found in sEVs can be transferred between cells by sEVs and induce inflammation more potently than naked mtDNA, as it is partially protected from DNase digestion.

Recently, an article by Neuberger et al. demonstrated and characterized the kinetics and the localization of DNA in association with EV through a physiological model of exercise [38]. The authors isolated EVs from the plasma of healthy humans by size exclusion chromatography or immuno-affinity capture followed by magnetic separation for EV markers: CD9^+^, CD63^+^ and CD81^+^. DNA was quantified without or with the addition of DNase. They have shown that only a small part of the DNA is protected inside the EV and that more DNA is associated with the surface of the EV. Compared to an article by Jeppesen et al. (2019), Neuberger et al. (2021) used a different method of isolating than UC. It is therefore not known whether the authors analyzed DNA in exosomes, microvesicles, large particles or non-exosomal compartments. However, as we followed the protocol by Jeppesen et al. (2019), we confirmed that approximately 35% of ecDNA is located inside sEVs. The rest of the DNA should be localized on the surface of sEVs, which is in coherence with the results of Neuberger et al. (2021) and Lázaro-Ibáñez et al. (2019). Furthermore, our results show that the portion corresponding to ecDNA inside sEVs is protected from degradation by DNase.

Relatively little information is known about the biological and physiological properties of EV-associated DNA, such as the mechanism responsible for DNA release via vesicles or the relative contribution of individual cell types to the total EV-associated ecDNA population. sEVs are involved in maintaining cellular homeostasis by removing fragmented DNA from cells by exocytosis. It is a physiological process. The inhibition of exocytosis would lead to intracellular accumulation of cytosolic DNA, which would lead to the activation of DNA damage dependent on reactive oxygen species. It would be followed by cell aging and, eventually, cell death. The sustained release of DNA by vesicles by aging cells has been observed. This suggests that cytosolic DNA fragments could be a significant source of intracellular stress associated with increased exosome secretion [39]. EcDNA protected from degradation could have other harmful effects during various diseases, such as prostate cancer [40]. The transfer of DNA via EVs can have the ability to influence the function of the recipient cells by increasing DNA-coding mRNA and the concentration of proteins. The methylation of EV-associated DNA can serve as a biomarker for the detection of cancer. The similarities in the methylation of genomic DNA and EV-associated DNA from the plasma, serum or saliva have been observed [41]. EVs have a role in horizontal DNA gene transfer. In this way, they can transfer pre-existing mutations in DNA, which has a relevant role in cancer progression, cell evolution and development [42], or they can be potential biomarkers of cancer and atherosclerosis [43]. Given that potential clinical samples are rarely processed immediately and are often frozen, one of the possible advantages of EV-associated DNA is that this DNA is stable under different storage conditions [44].

The amount of ecDNA per one exosome was recalculated based on the particle concentration measured by Nanosight (which was in the order of 10^12^) and the average ecDNA concentration in sEVs. As a result, there is 10^−12^ ng of ecDNA per sEV, which is less than the weight of one copy of DNA. We therefore conclude that since less than one copy is present in one sEV, ecDNA is not present in every vesicle.

In conclusion, in this study, we managed to confirm the presence of ecDNA in sEVs isolated from human plasma by the golden standard also employed by Jeppesen et al. (2019). Although the author claims the opposite, using the same method, we demonstrated that DNA is associated with sEVs, and it seems that most of it is mtDNA rather than ncDNA. A higher concentration of ecDNA is on the surface of sEVs; therefore, 35% is protected from DNase. Further studies should focus on the potential harmful autoimmune effect of DNA carried by EVs or specifically sEVs.

## 4. Materials and Methods

### 4.1. Plasma Sampling

Blood was collected from healthy volunteers from the Institute of Molecular Biomedicine into heparin tubes (Greiner Bio-One Plasma Tubes with Lithium Heparin and Gel, Greiner Bio-One GmBH, Kremsmünster, Austria). Three heparin tubes were taken from one volunteer in a total volume of approximately 30 mL of blood. The processing of samples was initiated within 30 min of drawing blood. Plasma was isolated from the collected blood by differential centrifugation. The blood was centrifuged at 1600× *g* for 10 min. Subsequently, the supernatant was pipetted into 2 mL tubes (Eppendorf, Germany) and centrifuged at 16,000× *g* for 10 min to remove apoptotic bodies and blood cell debris [45]. One ml of plasma was kept, and the rest was used to isolate sEVs—further centrifuged at 16,000× *g* for another 20 min (Figure 6). All centrifugation steps were performed at 4 °C. After the centrifugations, the plasma was divided into two groups of the same volume: the first group was treated with TURBO™ DNase (TURBO DNA-free™ Kit, Thermo Fisher Scientific, Waltham, MA, USA) according to the manufacturer’s instructions. TURBO™ DNase (2 Units/µL) was added to plasma at a volume of 3 µL. The second group was not treated with TURBO™ DNase (Figure 7).

### 4.2. Isolation of sEVs from Plasma Samples

Plasma with or without DNase dedicated for sEVs isolation was further processed by differential UC (Figure 6). Briefly, the first UC was at 100,000× *g* for 90 min in a Type 70.1 Ti Fixed-Angle Titanium Rotor (Beckman Coulter Optima L-90K Ultracentrifuge, Beckman Coulter, Inc., Brea, CA, USA). After UC, the supernatant was discarded, and the pellet was resuspended in 90 μL phosphate buffer saline (PBS) (3 × 30 μL). The second UC was with discontinuous (12–36%) iodixanol-density gradient that was prepared using a commercial OptitPrep solution (Sigma-Aldrich, St. Louis, MO, USA). The collected pellets were placed at the bottom of an Open-Top Thinwall Ultra-Clear Tube (Beckman Coulter, Inc., Brea, CA, USA), and a solution of a descending concentration of the iodixanol density gradient was layered on top [20]. The samples were ultracentrifuged at 120,000× *g* for 15 h in an SW 32.1 Ti Swinging-Bucket Rotor (Beckman Coulter Optima L-90K Ultracentrifuge, Beckman Coulter, Inc., Brea, CA, USA) [20]. After UC, the upper fractions of the gradient were collected at a volume of 8 mL, pipetted into fusible ultracentrifuge tubes and supplemented with PBS solution. The third UC was at 120,000× *g* for 4 h in a Type 70.1 Ti Fixed-Angle Titanium Rotor. The resulting pellet was collected by pipetting in 90 μL PBS (3 × 30 μL PBS). All UC steps were performed at 4 °C.

### 4.3. Quantification of sEVs Proteins

The proteins were lysed in 20 mmol Tris-HCl solution and isolated from the sEVs using an ultrasonicator (Elmasonic P, Elma, Germany). The protein concentration was measured with a commercial BCA Protein Assay Kit (Thermo Fischer Scientific, Waltham, MA, USA).

### 4.4. Nanoparticle Tracking Analysis

Nanoparticle tracking analysis (NTA) was performed using a Nanosight NS500 instrument (Malvern Panalytical Ltd., Malvern, UK). The particles were 1000× diluted in PBS solution. The temperature during the measurement was 22 °C based on the NTA Temperature program. NTA recorded 30 s videos of each measurement of the sample by a CCD camera.

### 4.5. Western Blot

sEVs markers were detected by western blot. A 10% SDS polyacrylamide gel was used. The samples were mixed with bromophenol blue dye at a 3:1 ratio, and 25 μL of the prepared samples was pipetted into one well. Gel electrophoresis lasted 30–45 min at 90 V and 60 min at 120 V. Subsequently, the proteins from the gel were blotted onto a polyvinylidene fluoride membrane (Bio-Rad Laboratories, Inc., Hercules, CA, USA) in a Turbo-Blot Turbo Transfer System (Bio Rad Laboratories, Inc., Hercules, CA, USA). After blotting the proteins, the membrane was blocked overnight in 5% bovine serum albumin (BSA) solution (Sigma-Aldrich, St. Louis, MO, USA) to prevent the non-specific binding of primary antibodies. The primary antibodies used to detect the proteins were 25 kDa anti-CD9, 45 kDa anti-TSG101 and 91 kDa anti-Calnexin (as a negative control) (Abcam, Inc., Cambridge, UK). All antibodies were diluted 1:1000 in 5% BSA solution. The primary antibody membrane was incubated for one hour at room temperature. After incubation, the membrane was washed three times in PBS for 5 min and then incubated with goat anti-mouse IgG and goat anti-rabbit IgG secondary antibodies (Abcam, UK) for one hour at room temperature. Secondary antibodies were diluted 1:2000 in 5% BSA solution. After incubation, the proteins were visualized using an LI-COR Odyssey instrument (BioAgilytix Labs, Hamburg, Germany).

### 4.6. Transmission Electron Microscopy

sEVs in PBS were visualized using a Morgagni 268D transmission electron microscope (FEI, Brno, Czech) at a magnification of 56,000–71,000×, which was equipped with a MegaViewIII digital camera (Soft Imaging System). This measurement was performed at the Institute of Histology and Embryology—Faculty of Medicine, Masaryk University in Brno. Thawed sEVs in PBS were used for visualization and placed on a Formvar/carbon-coated copper grid (HF35Cu, PyserSGI Limited, Kent, UK). sEVs were placed on the grid for two minutes at room temperature. Subsequently, the grid was rinsed in redistilled water and stained with 2% ammonium molybdate 1:1 at room temperature for 20–30 s. The dye was then aspirated through filter paper. The samples were immediately observed under a microscope.

### 4.7. Flow Cytometry

Flow cytometry (FC) was performed using a Dx FLEX Flow Cytometer (Beckman Coulter Inc., Brea, CA, USA). FC was pre-calibrated using calibration beads of sizes of 100 nm, 200 nm, 500 nm and 1000 nm. The gating strategy was used as follows: all events were plotted using a V-SSC-H channel on an *x*-axis and an FITC-A channel on a *y*-axis. Signal noise was excluded based on the Flow Cytometry Sub-Micron Sizer Reference Kit and Green Fluorescent beads (cat. Number F13839, Thermo Fisher, Waltham, MA, USA) by the selection of events with fluorescence ≥100 nm FITC-labeled beads. Particular size gates were then set up according to 100 nm, 200 nm and 500 nm bead populations and further verified by creating a calibration curve from the V-SSC-H median of the corresponding population. The high- and low-fluorescent vesicles population was gated based on the profile displayed on a representative plot (Figure 8). The samples were diluted in filtered 0.9% NaCl and divided into aliquots. All aliquots were stained with 2 µM Sytox Green Nucleic Acid Stain (Invitrogen, Waltham, MA, USA). The first aliquot was left as the control, and the second was treated first with TURBO™ DNase (2 U/μL) (TURBO DNA-free™ Kit, Thermo Fisher Scientific, Waltham, MA, USA) according to the manufacturer’s recommendation. All samples were incubated for 30 min in the dark at room temperature. The samples were excited by a 488 nm laser, collected on a 525/40 BP filter and measured for one minute at a slow flow rate (10 μL/min).

### 4.8. Analysis of ecDNA

EcDNA was isolated from 200 μL of plasma and from approximately 90 μL of sEVs using a commercial QIAamp DNA Mini Kit (Qiagen, Hilden, Germany). according to the instructions of a manufacturer. It was three types of sEVs: sEVs, Dplas-sEVs and D-sEVs (Figure 2). The concentration of isolated ecDNA was measured fluorometrically using the Qubit^®^ 1X dsDNA HS Assay Kit (Thermo Fisher Scientific, Waltham, MA, USA) and a Qubit^®^ 3.0 fluorometer (Thermo Fisher Scientific, Waltham, MA, USA). The resulting concentration of DNA was recalculated to 1 mL of plasma.

The subcellular origin of isolated ecDNA was estimated by qPCR. SsoAdvanced universal SYBR Green Supermix (Bio-Rad, Hercules, CA, USA) was used. The β-globin primers were used to quantify human ncDNA (F: GCT TCT GAC ACA ACT GTG TTC, R: CAC CAA CTT CAT CCA CGT TCA). The D-loop primers were used to quantify human mtDNA (F: CAT AAA AAC CCA ATC CAC ATC A, R: GAG GGG TGG CTT TGG AGT). The PCR program consisted of an initiation lasting 3 min at 98 °C; 40 cycles of 15 s at 98 °C, 30 s at 51 °C for ncDNA and 47 °C for mtDNA and 30 s at 60 °C, followed by the melting curve. The program was run in a MasterCycler RealPlex 4 thermal cycler (Eppendorf, Hamburg, Germany).

### 4.9. Position of ecDNA in sEVs

sEVs were divided into two groups and pipetted into a black 96-well plate to determine the position of the ecDNA. One group of sEVs was treated with 0.05% Triton X-100 (Sigma-Aldrich, St. Louis, MO, USA), which disrupted the vesicular membrane, thereby releasing the DNA into extracellular space. The second group of sEVs remained intact. Sytox Green Nucleic Acid Stain (0.05 μM) (Thermo Fisher Scientific, Waltham, MA, USA) was added to both samples. The prepared samples were incubated at 37 °C for 30 min. After incubation, fluorescence was measured on a Synergy 4 spectrophotometer (BioTek, Winooski, VT, USA) at an excitation wavelength of 488 nm and an emission wavelength of 525 nm.

### 4.10. Statistical Analysis

All statistical analyses and graphs were performed using Graph Pad Prism (version 8 or higher, San Diego, CA, USA) and a *t*-test. Data are shown as the mean ± SEM. Significance for all statistical analyses was assumed at *p* < 0.05.

## Figures and Tables

**Figure 1 ijms-24-05915-f001:**
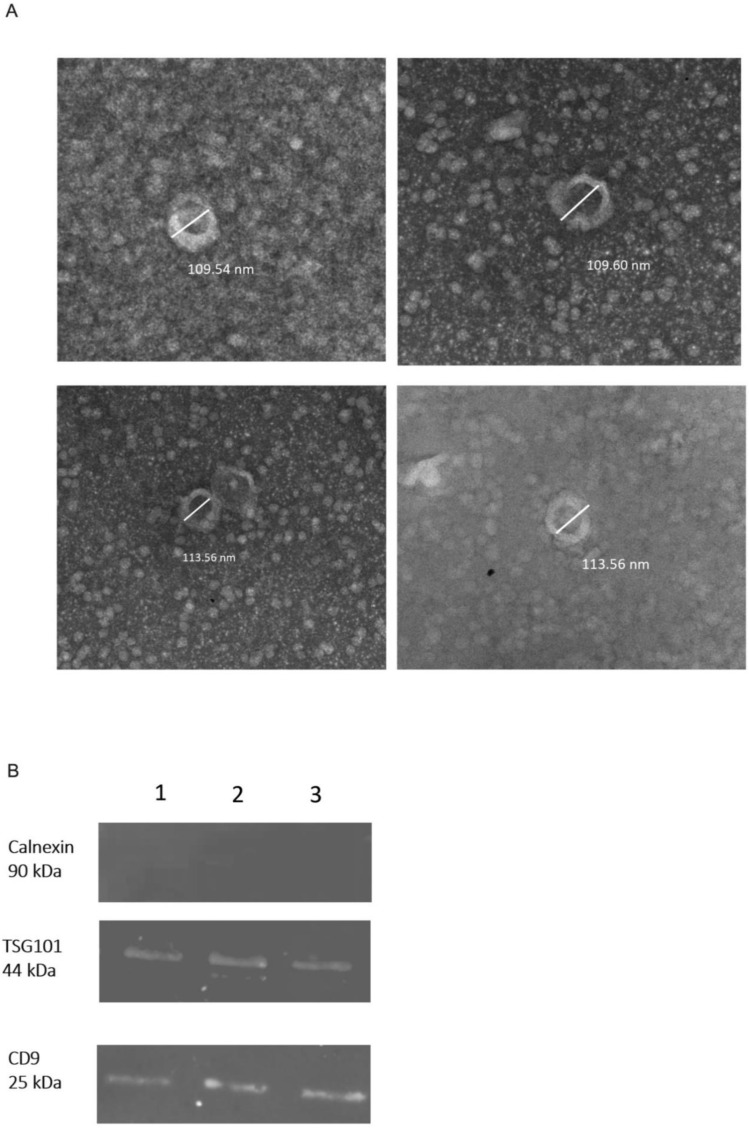
Confirmation of sEVs. (**A**) Negative stained sEVs isolated from human plasma observed by a transmission electron microscope; (**B**) Western blot analysis for sEVs markers in three separate sEVs isolations [1,2,3].

**Figure 2 ijms-24-05915-f002:**
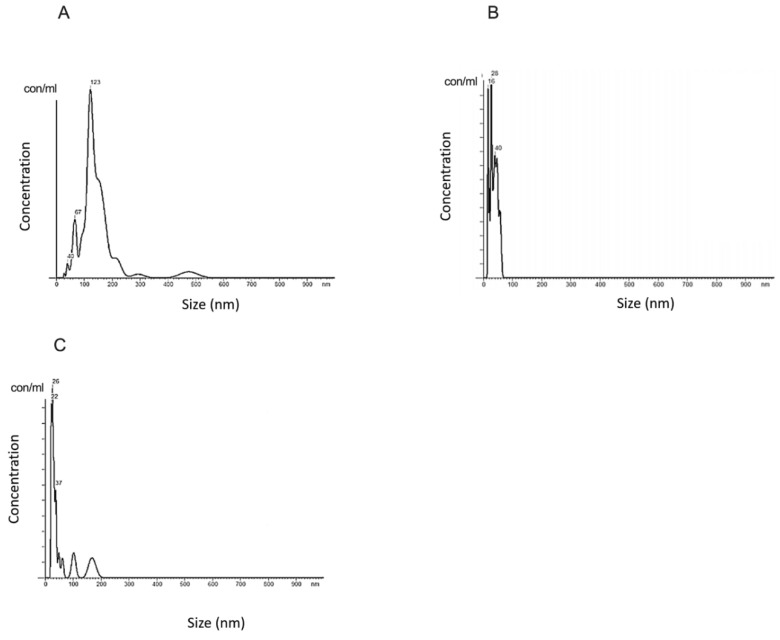
Nanoparticle tracking analysis of sEVs. (**A**) sEVs isolated from fresh human plasma without DNase; (**B**) Dplas-sEVs–sEVs from DNase-treated plasma; (**C**) D-sEVs–sEVs treated with DNase.

**Figure 3 ijms-24-05915-f003:**
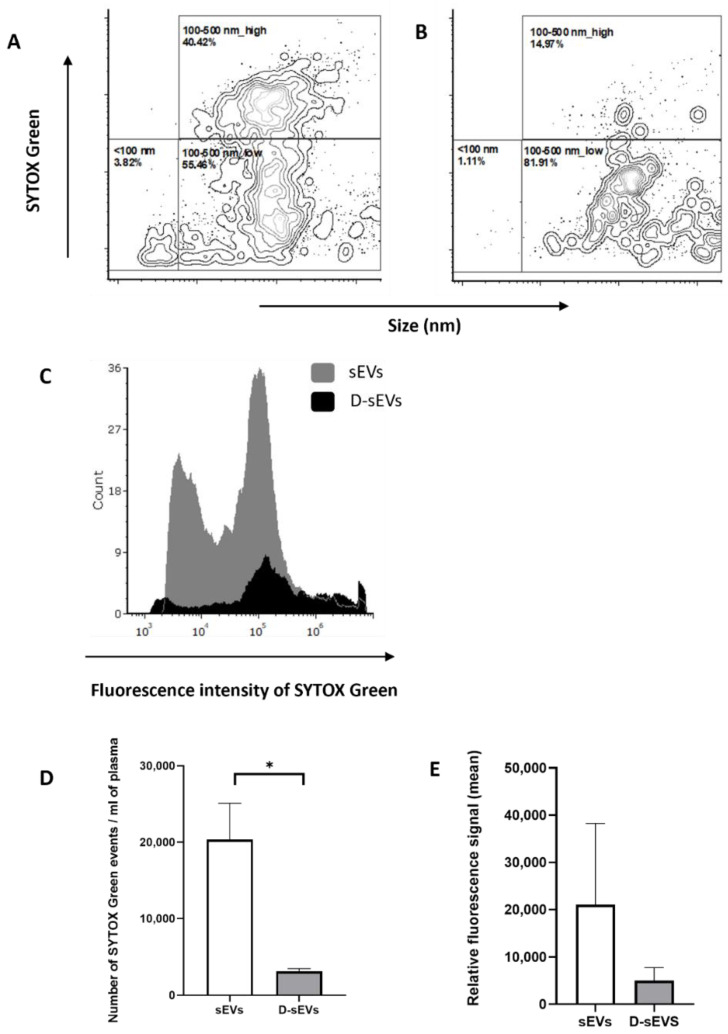
Flow cytometry of sEVs. (**A**) sEVs isolated from fresh human plasma without DNase; (**B**) Dplas-sEVs–sEVs isolated from fresh human plasma with the addition of DNase; (**C**) overlay histogram of sEVs isolated from fresh human plasma with and without DNase; (**D**) number of detected events per ml of plasma; (**E**) relative fluorescence signal of ecDNA stained with SytoxGreen. Data are shown as the mean ± SEM. Significance was assumed at *p* < 0.05. * *p* < 0.05. sEVs–EVs isolated from fresh human plasma without DNase; D-sEVs–sEVs treated with DNase.

**Figure 4 ijms-24-05915-f004:**
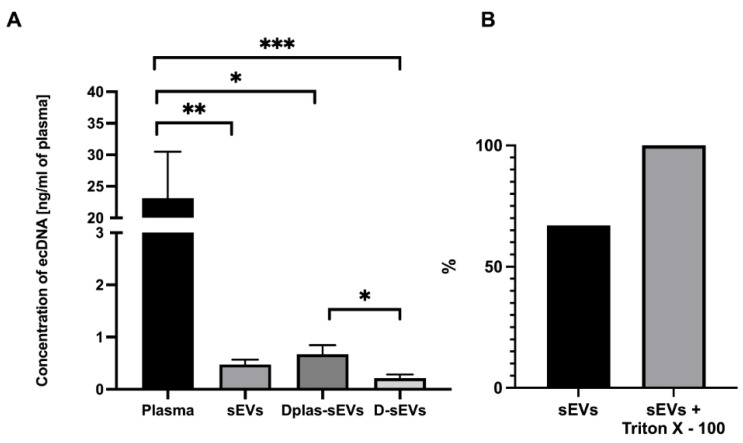
EcDNA in plasma and sEVs. (**A**) Quantification of ecDNA in plasma and sEVs. (**B**) Percentage of ecDNA outside of sEVs. Percentage shows the average concentration of DNA. sEVs–sEVs isolated from non-treated plasma, Dplas-sEVs–sEVs isolated from DNase-treated plasma and D-sEVs–sEVs treated with DNase. * *p* ≤ 0.05; ** *p* ≤ 0.01; *** *p* ≤ 0.001; data are shown as the mean ± SEM.

**Figure 5 ijms-24-05915-f005:**
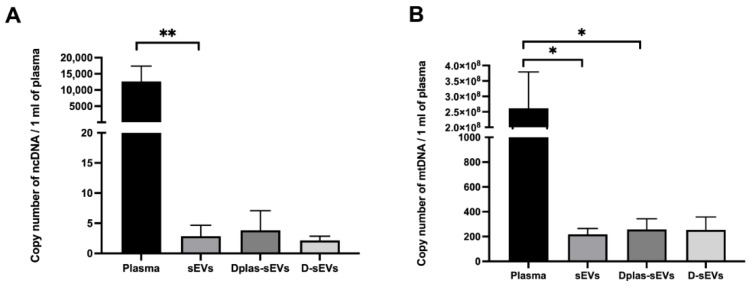
Number of DNA copies/mL of plasma in plasma and sEVs. (**A**) Nuclear DNA; (**B**) mitochondrial DNA. sEVs–exosomes isolated from DNase non-treated plasma. Dplas-sEVs–sEVs isolated from DNase-treated plasma. D-sEVs–sEVs treated with DNase. * *p* ≤ 0.05; ** *p* < 0.01.

**Figure 6 ijms-24-05915-f006:**
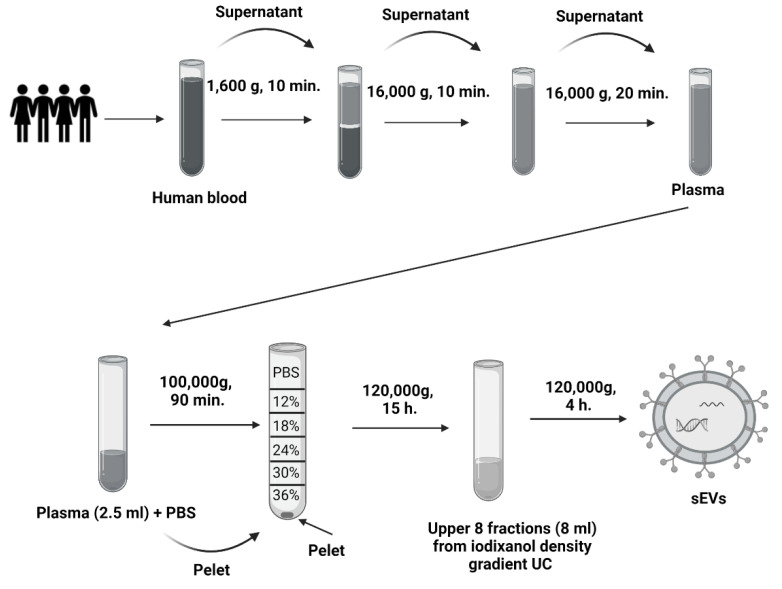
Scheme of sEVs isolation. Scheme of the centrifugation of human plasma and differential ultracentrifugation with iodixanol density gradient.

**Figure 7 ijms-24-05915-f007:**
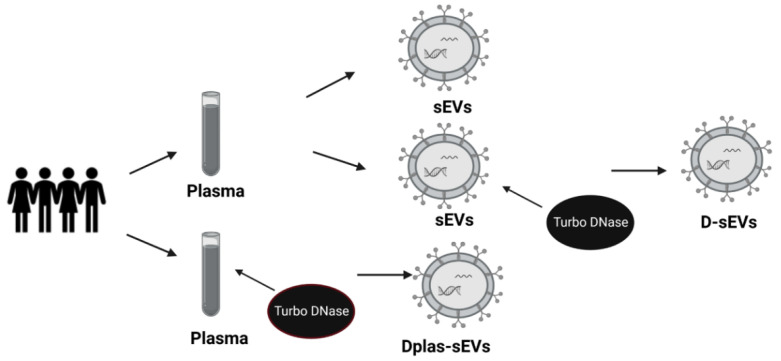
Scheme of sEVs samples. sEVs—small extracellular vesicles from DNase-free plasma; Dplas-sEVs—sEVs from Dnase–treated plasma; D-sEVs—sEVs treated with Dnase.

**Figure 8 ijms-24-05915-f008:**
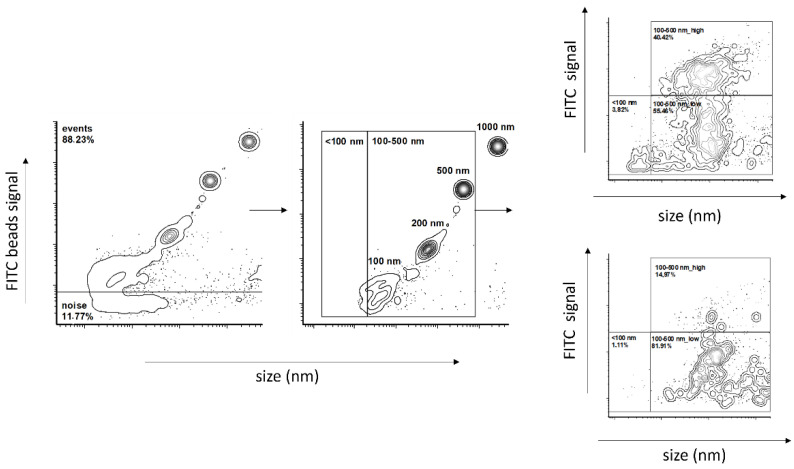
Flow cytometry gating strategy.

## Data Availability

No new data were created or analyzed in this study. Data sharing is not applicable to this article.

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
