# Peer review of "Investigation of the Presence of DNA in Human Blood Plasma Small Extracellular Vesicles"

_ijms, 2023, doi:10.3390/ijms24065915_

Round 1
Reviewer 1 Report
Dear Authors,
The paper entitled “Investigation of the presence of DNA in human blood plasma small extracellular vesicles" submitted for publication in International Journal of Molecular Sciences represents high scientific work, and it can be accepted for publication in International Journal of Molecular Sciences after minor revision.
As authors wrote “The aim of this study was to confirm the presence of ecDNA in sEVs isolated from human plasma using protocol proposed by Jeppesen et al., to eliminate non-EVs compartments”.
In this work authors show the localization and subcellular origin of ecDNA associated with sEVs and estimate their approximate concentration.
Article is well written, with clear hypothesis and well described materials and methods paragraph.
In discussion authors interestingly present and compare their results with newest original literature in this field of science.
Minor Comments:
please expand abbreviation the first time they are mentioned in the article (eg. Line 88, 91, and others)
Introduction
Line 41- please add reference for EVs size classification
Results- correct the figure numerations. The first figure in this article has number 3
The tile of each figure should be added, please clarify it.
Expand the abbreviations in all figures descriptions.
Figure 5- please add the gating strategy with calibration beads
Fig 5C- please add fluorescence intensity of SYTOX Green in x axis
Fig 5D- axis Y- please add number of SYTOX Greene events in y axis
Fig 6.B- near the bar graph is 2.4
Materials and methods
Line 327- add reference for this centrifugation protocol
Paragraph 4.9 line440-447- change italic style
Discussion
Is there any possibility to know the origin of the sEVs? In my opinion, this aspect of the cell origin of plasma EVs should be discussed.
Author Response
Minor Comments:
1. please expand abbreviation the first time they are mentioned in the article (eg. Line 88, 91, and others)
Thank you for this comment. We made the corrections.
Introduction
2. Line 41- please add reference for EVs size classification
We added the reference.
3. Results- correct the figure numerations. The first figure in this article has number 3
The tile of each figure should be added, please clarify it.
Expand the abbreviations in all figures descriptions.
Thank you for the notice. We corrected all figure numerations and added the titles. We added the explanations of abbreviations to the figure legends.
4. Figure 5- please add the gating strategy with calibration beads
Fig 5C- please add fluorescence intensity of SYTOX Green in x axis
Fig 5D- axis Y- please add number of SYTOX Greene events in y axis
Regarding Figure 5, we made all corrections and we added the gating strategy (Fig. 8)
5. Fig 6.B- near the bar graph is 2.4
We made the correction.
6. Materials and methods
Line 327- add reference for this centrifugation protocol
Paragraph 4.9 line440-447- change italic style
We made the corrections.
7. Discussion
Is there any possibility to know the origin of the sEVs? In my opinion, this aspect of the cell origin of plasma EVs should be discussed.
We thank the reviewer for the interesting suggestion. The origin of EVs was not the aim of the study and we have not done any experiments giving us such results. Therefore we did not dare to have any opinion on this in the discussion. However, for the future studies, the origin of EVs carrying DNA is certainly an interesting question to be answered. Due to the lack of samples used in our study as well as the lack of the time, we did not add anything on this topic to the experimental part, therefore also not to the discussion.
Reviewer 2 Report
In this paper, the authors present evidence for the presence of DNA in sEVs (~120nm in size and carrying CD9 and TSG101)) isolated from human plasma. The authors report that 60-75% of DNA was associated with sEV surface, but a part of DNA (~35%) was inside the vesicles, as shown by Triton X treatment of the vesicles. The DNA associated with sEVs was of nuclear and mitochondrial origin, but most appears to be mitochondrial. sEVs contained about 2% of ecDNA present in plasma. Following DNase treatment of sEVs, their size, number and intensity of the fluorescent signal decreased. The concentration of ecDNA was significantly lower in sEVs treated with DNase than in sEVs isolated from DNase treated plasma.The authors then discuss the biological role of sEV associated DNA, suggesting that it might have potentially harmful effects by inducing inflammation.
The authors suggest that DNase treatment of sEVs dissociates sEV aggregates possibly formed by histones interacting with phosphatidylinositol -containing vesicles. and resulting in smaller vesicles. When plasma was treated with DNase before ultracentrifugation, the formation of aggregates was prevented, giving vesicles with a smaller size. These data suggest that DNA located on the surface of vesicles may lead to their aggregation.
In view of the existing controversy in respect to DNA in sEVs, this article is an important contribution. The experiments are rigorously performed, the study design involving comparisons of sEVs from plasma pre-treated with DNase vs DNase treated sEVs the well done vesicle isolation and characterization, all contribute to the significance of the presented data. The manuscript is well written, and the data presentation is satisfactory.
Author Response
We would like to thank to the reviewer for the comments.
Reviewer 3 Report
Based on the contradictory literature about the presence of DNA in small extracellular vesicles (sEVs), Lichá and colleagues aims to confirm the presence of extracellular DNA (ecDNA) in sEVs isolated from fresh human plasma based on previous published methods. As final goal, the study should clearly investigate the localization and subcellular origin of the ecDNA associated with sEVs as well as the estimation of the approximate concentration.
At this state, the article requires a considerable amount of work. It is necessary to improve it in order to make it more understandable:
· The mean of certain acronyms should be carefully added in each part. For instance, EcDNA is not explained in the abstract, but only in the introduction. Dplas-sEVs meaning is in the materials and methods and not in the results. Each part should have the given acronym.
· Conceptual meaning of “sEVs from DNase-free plasma (sEVs)”, “sEVs from DNase-treated plasma (Dplas-sEVs)” and “sEVs treated with DNase (D-sEVs)” should be formally explained also in the rrsults, being this key samples and concepts used in the study. This would considerably improve paragraph 2.1., currently a little difficult to follow.
· The order of figures should represent their position. Figure 1 cannot be at the end of the manuscript. That could/should be “Supplementary Figure 1”.
· Line 80, 81, … When a figure is mentioned in the text, the letters should be applied. Example, electron microscopy is Fig. 3A, western blot (WB) Fig. 3B etc…
· Line 84. Not clear the part of calnexin. Aren’t the data in Figure 3B?
· Since the manuscript aims to a deep sEVs characterization, there should be more markers in the WB, including CD63, CD81 and ALIX, and a further negative marker. Furthermore, this should be done on all sources/treated sEV. Finally, the same markers should be showed in the cell of origin, peripheral blood mononuclear cells (PBMCs) in this case.
Few experimental things are not clear:
· Small EV isolation requires more attention depending on the sources. Here the author uses plasma, notoriously much more complex than cell supernatant. Usually the sample “clearing” process should be more delicate. A centrifugation of 16.000xg (even for few minutes) would objectively break the cells which are being pulled down, leading to release of intracellular material – above all when in 2mL of volume. I would highly suggest a more gentle 2000xg (40-60 min), followed by at least 10000xg (60 min) Good the UC part.
· About the different sEVs size comparisons. Currently, no nano size particle machine can be reliable to measure biological samples below 30nm. Indeed, despite what certain machines claim, only synthetic particles can be decently measures if smaller than 30nm. Based on this, a far create amount of <30nm sEVs represent a sEVs enrichment, or simply a greater amount of background wrogly recognized as sEVs by the machine?
Novelty
· The scope of the manuscript, is to truly demonstrate whether ecDNA is in sEVs. Above all, compared with Jeppesen et Al. the author is confident of the results given the fresh samples used. The strategy to achieve such goal should be well thought. First, I would say that only 3 control plasma samples are no where near to express a solid concept. The experimental n should be increased. Furthermore, there should be a parallel correlation with sEVs isolated from cell supernatant. A different of ecDNA amount/localization/presence in sEVs depending on the source would represent a much greater impact for the community.
Overall, ecDNA presence or nor in sEVs has been largely already investigated. I strongly agree with the author that the literature is contradictory and that presence of absence of DNA in sEVs can be due to technical pitfalls or different vesicles sources. For this reason, being the aim of the manuscript to confirm something so discussed, the author should extend its research to comparing difference sEV sources (as mentioned above) as well as different isolation techniques, what about size exclusion chromatography (SEC)?
Author Response
1. The mean of certain acronyms should be carefully added in each part. For instance, EcDNA is not explained in the abstract, but only in the introduction. Dplas-sEVs meaning is in the materials and methods and not in the results. Each part should have the given acronym.
We explained ecDNA in the abstract. We corrected all the abbreviations – they are explained first time they are mentioned in the text.
2. Conceptual meaning of “sEVs from DNase-free plasma (sEVs)”, “sEVs from DNase-treated plasma (Dplas-sEVs)” and “sEVs treated with DNase (D-sEVs)” should be formally explained also in the rrsults, being this key samples and concepts used in the study. This would considerably improve paragraph 2.1., currently a little difficult to follow.
We apologize for the confusion. We explained the abbreviations in the results.
3. The order of figures should represent their position. Figure 1 cannot be at the end of the manuscript. That could/should be “Supplementary Figure 1”.
We apologize for the wrong order of the figures. The error must have happened during the submission process. We corrected it.
4. Line 80, 81, … When a figure is mentioned in the text, the letters should be applied. Example, electron microscopy is Fig. 3A, western blot (WB) Fig. 3B etc…
We applied the letters into the text.
5. Line 84. Not clear the part of calnexin. Aren’tthe data in Figure 3B?
We apologize for the confusion. We deleted “data not shown”. We hope it is clear now.
6. Since the manuscript aims to a deep sEVs characterization, there should be more markers in the WB, including CD63, CD81 and ALIX, and a further negative marker. Furthermore, this should be done on all sources/treated sEV. Finally, the same markers should be showed in the cell of origin, peripheral blood mononuclear cells (PBMCs) in this case.
The extracellular vesicles were isolated by standard method – differential ultracentrifugation followed by density gradient purification of EVs which enables the preparation of EVs with high quality. These EVs were subsequently characterized by different independent methods to meet the minimal criteria of the ISEV community and to confirm that the sample we work with is sEVs. We believe that the confirmation by Western blotting with markers commonly used in published studies, transmission electron microscopy, and nanoparticle tracking analysis are sufficient. We are aware that we did not confirm the presence of other existing markers, however, due to low yield of the samples, it was not possible. There are many studies already published using only a few chosen markers (some examples of the studies are below). We believe that confirmation of chosen exosomal markers along with other methods confirming the good isolation is sufficient for this study. According to MISEV2018, it should be sufficient.
PBMCs are only one type of cells in blood therefore the cell of origin was not used in this study alongside with vesicles isolated from blood plasma. Unfortunately, we could not keep the blood from donors used in this study, therefore we no longer have any kind of cells from the donors and we cannot perform additional experiments. However, we will do it in our next study.
Helwa, I., Cai, J., Drewry, M. D., Zimmerman, A., Dinkins, M. B., Khaled, M. L., Seremwe, M., Dismuke, W. M., Bieberich, E., Stamer, W. D., Hamrick, M. W., & Liu, Y. (2017). A comparative study of serum exosome isolation using differential ultracentrifugation and three commercial reagents. PLoS ONE, 12(1). https://doi.org/10.1371/journal.pone.0170628
Kharmate, G., Hosseini-Beheshti, E., Caradec, J., Chin, M. Y., & Tomlinson Guns, E. S. (2016). Epidermal growth factor receptor in prostate cancer derived exosomes. PLoS ONE, 11(5). https://doi.org/10.1371/journal.pone.0154967
Yokoi, A., Villar-Prados, A., Oliphint, P. A., Zhang, J., Song, X., DeHoff, P., Morey, R., Liu, J., Roszik, J., Clise-Dwyer, K., Burks, J. K., O’Halloran, T. J., Laurent, L. C., & Sood, A. K. (2019). Mechanisms of nuclear content loading to exosomes. Science Advances, 5(11), 1–17. https://doi.org/10.1126/sciadv.aax8849
Zhang, Y., Song, K., Qi, G., Yan, R., Yang, Y., Li, Y., Wang, S., Bai, Z., & Ge, R. li. (2020). Adipose-derived exosomal miR-210/92a cluster inhibits adipose browning via the FGFR-1 signaling pathway in high-altitude hypoxia. Scientific Reports, 10(1). https://doi.org/10.1038/s41598-020-71345-8
Few experimental things are not clear:
7. Small EV isolation requires more attention depending on the sources. Here the author uses plasma, notoriously much more complex than cell supernatant. Usually the sample “clearing” process should be more delicate. A centrifugation of 16.000xg (even for fewminutes) would objectively break the cells which are being pulled down, leading to release of intracellular material – above all when in 2mL of volume. I would highly suggest a more gentle 2000xg (40-60 min), followed by at least 10000xg (60 min) Good the UC part.
We would like to thank to the reviewer for the suggestions. In our study, we decided to follow the step-by-step protocol by Jeppesen et al. study to exclude the potential isolation of non-exosomal vesicles containing DNA. The first, step after the blood draw was the centrifugation step at 1600g to separate cells from plasma. Followed by 16000 g to remove apoptotic bodies and blood cell debris and the next 16 000 for 20 min. Subsequently, the samples of plasma were centrifugated at 100 000 g and purified using density gradient protocol. Thus, we believe that we obtained EVs of appropriate quality by this protocol commonly used in published studies. Also the protocol ensures no cells breakage.
8. About the different sEVs size comparisons. Currently, no nano size particle machine can be reliable to measure biological samples below 30nm. Indeed, despite what certain machines claim, only synthetic particles can be decently measures if smaller than 30nm. Based on this, a far create amount of <30nm sEVs represent a sEVs enrichment, or simply a greater amount of background wrogly recognized as sEVs by the machine?
We thank the reviewer for this important comment. The official webpage of Nanosight machine says that the NS500 provides detailed and intuitive analysis of the size distribution and concentration of all types of nanoparticles from 10nm to 2000nm in diameter, depending on the instrument configuration and sample type being analyzed. We agree that such sample as used in our study could be difficult to analyze by Nanosight. We added a comment to the discussion.
However, still after usage of DNase – either directly into the plasma or into sEVs sample – we see the change regarding the size profile of vesicles, specifically larger vesicle concentrations decreased.
Novelty
9. The scope of the manuscript, isto truly demonstrate whether ecDNA is in sEVs. Above all, compared with Jeppesen et Al. the author is confident of the results given the fresh samples used. The strategy to achieve such goal should be well thought. First, I would say that only 3 control plasma samples are no where near to express a solid concept. The experimental n should be increased. Furthermore, there should be a parallel correlation with sEVs isolated from cell supernatant. A different of ecDNA amount/localization/presence in sEVs depending on the source would represent a much greater impact for the community.
We would like to explain it more clearly. The scope of the manuscript was to demonstrate whether ecDNA is in sEVs isolated from plasma of healthy volunteers in comparison to many other studies where cell lines are used. Since it is the basic research we chose to work with the low number of samples. Blood donated by volunteers was withdrawn multiple times since the yield of sEVs is low. First we used sEVs for confirmation that we truly worked with vesicles – electron microscopy and NTA. Several further donations were used to obtain a sufficient amount of proteins to perform western blot and confirm the presence of exosomal markers. The yield of proteins quantified from isolated sEVs of plasma was also low, therefore we chose the markers mentioned in the study only. Further donations and isolations of sEVs were used to isolate DNA only and ensure that the yield is sufficient for both fluorometry and PCR. We would like to add that especially for DNA quantification, we performed the whole experiment three times with the same three donors to ensure the results could be likely. Last but not least, one more isolation from donated blood were used for flow cytometry analysis.
10. Overall, ecDNA presence or nor in sEVs has been largely already investigated. I strongly agree with the author that the literature is contradictory and that presence of absence of DNA in sEVs can be due to technical pitfalls or different vesicles sources. For this reason, being the aim of the manuscript to confirm something so discussed, the author should extend its research to comparing difference sEV sources (as mentioned above) as well as different isolation techniques, what about size exclusion chromatography (SEC)?
We agree, that there are many different isolation techniques that could be used for EVs isolation. The selected protocol follows the protocol of Japessen et al., which seems to be reliable and represents a gold standard for EVs isolation. There are already many other studies comparing different isolation techniques and we already know that the choice of the isolation technique impacts the results. According to minimal criteria of the ISEV community, it is important to report all the isolation steps in detail, which we have done. In our study, we focused on EVs isolated from plasma as a potential biological material source in the clinic. We believe that our results bring relevant findings to the discussion about DNA presence in EVs. In the future we could consider isolation of sEVs from different sources, although the aim was to use plasma, which is of course more heterogeneous.
Round 2
Reviewer 3 Report
No further comments.